# Genetic Predisposition to SARS-CoV-2 Infection: Cytokine Polymorphism and Disease Transmission within Households

**DOI:** 10.3390/biology12111385

**Published:** 2023-10-30

**Authors:** Marius Saal, Henry Loeffler-Wirth, Thomas Gruenewald, Ilias Doxiadis, Claudia Lehmann

**Affiliations:** 1Laboratory for Transplantation Immunology, University Hospital Leipzig, Johannisallee 32, 04103 Leipzig, Germany; saal.marius@googlemail.com (M.S.); ilidox1@icloud.com (I.D.); 2Interdisciplinary Centre for Bioinformatics, IZBI, Leipzig University, Haertelstr. 16–18, 04107 Leipzig, Germany; wirth@izbi.uni-leipzig.de; 3Clinic for Infectious Diseases and Tropical Medicine, Klinikum Chemnitz, Flemmingstraße 2, 09116 Chemnitz, Germany; t.gruenewald@skc.de

**Keywords:** SARS-CoV-2, COVID-19, cytokine polymorphism, transmission, disease susceptibility, T helper cell subsets, pro-inflammatory cytokines, anti-inflammatory cytokines, Th17 cells, Treg cells

## Abstract

**Simple Summary:**

Soluble factors, the cytokines, influence the ability of an individual host defense against intruders, e.g., the SARS-CoV-2 virus. Distinct T cells are a major source of many of those molecules. The helper T cells are divided into categories according to their action on the immune response: the more pro-inflammatory helper T cells 1 (Th1: TNF-α, IFN-γ), the anti-inflammatory (Th2: IL-10) cells, and the more regulatory Th17 (IL-17, TGF-β1) and Treg (TGF-β1) cells. In the present report, we elaborate on the genetically determined activity of such cytokines, regarding defined polymorphisms with known impact on the cytokine expression and their influence on the course of SARS-CoV-2 infection. We selected from a larger cohort individuals from the same household (n = 58). We divided them into households with all individuals SARS-CoV-2-PCR positive (n = 29) with 61 individuals, mixed households (n = 24) with 62 individuals and households (n = 5) with 15 SARS-CoV-2-negative individuals and compared the frequency of distinct polymorphisms. The results obtained indicate a role for a genetically determined balance of the different T helper cell pathways.

**Abstract:**

We addressed the question of the influence of the molecular polymorphism of cytokines from different T helper subsets on the susceptibility to SARS-CoV-2 infection. From a cohort of 527 samples (collected from 26 May 2020 to 31 March 2022), we focused on individuals living in the same household (n = 58) with the SARS-CoV-2-infected person. We divided them into households with all individuals SARS-CoV-2 PCR positive (n = 29, households, 61 individuals), households with mixed PCR pattern (n = 24, 62) and negative households (n = 5, 15), respectively. TGF-β1 and IL-6 were the only cytokines tested with a significant difference between the cohorts. We observed a shift toward Th2 and the regulatory Th17 and Treg subset regulation for households with all members infected compared to those without infection. These data indicate that the genetically determined balance between the cytokines acting on different T helper cell subsets may play a pivotal role in transmission of and susceptibility to SARS-CoV-2 infection. Contacts infected by their index persons were more likely to highly express TGF-β1, indicating a reduced inflammatory response. Those not infected after contact had a polymorphism leading to a higher IL-6 expression. IL-6 acts in innate immunity, allergy and on the T helper cell differentiation, explaining the reduced susceptibility to SARS-CoV-2.

## 1. Introduction

In late 2019, the new SARS-CoV-2 virus quickly spread all over of the world within a matter of weeks. On 11 March 2020, the WHO declared a public health emergency of international concern [1]. The virus infects epithelial cells in the upper respiratory tract and can then migrate into the lung. Infection with the virus leads to a disease called COVID-19 (coronavirus disease 2019) with varying clinical courses from asymptomatic to severe. In severe cases, acute respiratory distress syndrome (ARDS) and complications such as acute kidney disease, coagulation disorders and thromboembolism may occur [2,3,4].

Although all age groups can be infected by the novel virus, the susceptibility for a symptomatic infection and the severity of the disease show a strong age dependency, making age an important risk factor for both infection and severe disease [5].

Risk factors for the severity of the disease are well studied and broadly understood [4,5,6]. Older age, one of the most important risk factors for severity, is characterized by significant immunological changes (immunosenescence) that can lead to chronic pro-inflammatory states. In addition, the prevalence of various comorbidities is high. Other demographic factors such as male sex are explained by lifestyle characteristics, different hormonal status, and a higher expression of the ACE-2 receptor on epithelial cells of the respiratory tract, which is crucial for SARS-CoV-2 invasion [6]. Other comorbidities leading to a higher risk of severe disease are hypertension, diabetes, cardiovascular diseases, and chronic lung diseases like COPD [4,6]. In addition, a wide range of laboratory indicators, such as HLA and blood groups, have been described as risk factors for severe disease and even function as predictive markers [7,8]. Severe COVID-19 goes along with significantly elevated levels of pro-inflammatory cytokines like IL-6, IL-8, IL-10, IL-2R, and TNF-α produced by a dysregulated immune system, leading to a pathophysiological phenomenon called “cytokine storm”, which is characterized by a rapid inflammatory cascade, resulting in hypercoagulability, tissue damage and multiorgan failure [6,9].

While risk factors for severe disease progression play a crucial role in the clinical treatment of COVID-19, risk factors for infection are still not well understood.

SARS-CoV-2 can spread quickly and frequently in so-called superclusters, meaning that a small number of infections lead to many secondary illnesses. The extent of the spread varies depending on the country and situation and can be influenced by barrier measures and other parameters [10,11]. The characteristics of the virus play a particularly important role here and can change over time due to growing population immunity, viral immune escape or selection pressure [11]. However, it is unclear why some people did not become infected with the virus, even if they were in close contact with an index case, such as living in the same household [12]. This leads to the hypothesis that there are factors that can make infection more or less likely.

In addition to age as a risk factor for infection with the virus, others have already been characterized and may play a role in the transmission of the virus. Male gender, pre-existing comorbidities such as hypertension, diabetes and cardiovascular diseases, ethnic disparities, immunological disorders, solid organ and stem cell transplantation and a profession as a health care worker are correlated with higher infection rates [6]. While the differences in infection rates in various professions and people in unequal social statuses can be explained by different exposure intensity and measures of protection [6], some immunological reasons are discussed for the other risk factors. Pre-existing comorbidities and older age may lead to immune compromise characterized by a chronic pro-inflammatory status. On the other hand, there are protective factors such as an adequate microbiome and a healthy diet, which reduce viral replication and the induction of pro-inflammatory cytokines during infection [6].

Already in 1989, Mosmann and Coffman [13] introduced the model for CD4+ cells as regulators of the adaptive immune system, with two subsets containing T-helper cells 1 (Th1) and T-helper cells 2 (Th2) with distinct cytokine profiles, and was widely supported [14]. The Th1 pathway leads to increased activity of CD8+ cytotoxic cells and macrophages, antibody-dependent cytotoxicity and secretion of opsonizing antibodies, all crucial for clearance of intracellular pathogens [14,15,16]. The Th2 pathway is important for mast cell proliferation and immunoglobulin production for the clearance of extracellular pathogens such as helminths as well as mucosal immunity [14,15,16]. The leading cytokines for the Th1 pathway are IFN-γ and TNF-α, while the Th2 cytokines are, e.g., IL-4, IL-5 and IL-10 [15,16]. These two pathways counteract and downregulate each other [14,15].

CD4+ cell differentiation is more diverse and distinct from the Th1 and Th2 lineages. Th17 and Treg cells are important players in the regulation and balanced orchestration of CD4+ [17]. While Th17 cells are pro-inflammatory in most circumstances and secrete IL-17 as well as other pro-inflammatory cytokines that recruit and target neutrophils to release IL-6 and IL-8, differentiation from naïve CD4+ cells is mediated by TGF-β1 and IL-6 [18]. Treg cells or T regulatory cells are modulators of the immune response and downregulate the effector cell response. They are characterized by the expression of CD4 and Foxp3. Cytokines differentially induce the development of these CD4+ subpopulations: TGF-β1 induces peripheral Treg cells and is able to induce a less-inflammatory Th17 subtype, whereas IL-1β induces the pro-inflammatory Th17 subtype [19]. Dysregulation of one of the pathways can lead autoimmune diseases [14,20] or secondary conditions, such as tissue damage [20]. 

Cytokines play a crucial role in the severity of the disease, and their expression level could be a risk factor for the susceptibility or intensity of inflammatory processes during the transmission and the course of disease. The pathogenesis of SARS-CoV-2 infection and the consequence of the cytokine storm were the reason for the focus in the present study on the role of cytokines in the susceptibility to SARS-CoV-2 infection [7,16]. It is known that disturbances of the IFN-γ pathway or the mitigation of IFN-γ activity, e.g., by antibodies, are risk factors for severe COVID-19 [21,22]. Tregs significantly influence the mode of action of helper T cells intervening in the secretion of the specific cytokines (reviewed by [23]). Mainly Th1 and pro-inflammatory Th17 cells are downregulated, and the release of TGF-β1 is promoted.

To explore the influence of various cytokine polymorphisms on the susceptibility to SARS-CoV-2, genotyping of important polymorphisms for cytokines involved [24,25] in the pathogenesis of COVID-19 was carried out. All five cytokines tested in this study were altered in previous studies in SARS-CoV-2-infected individuals [26,27,28,29,30,31,32].

In previous studies, the ratio of cytokine blood levels was considered a measure of immune activity [26], In light of these observations that genetic polymorphism and cytokine release potential affect cytokine release [33], this could give information on how the heterogeneous composition of cytokine genetics in individuals influence infection rates of SARS-CoV-2.

## 2. Materials and Methods

### 2.1. Cohort

Previously, a cohort of 527 samples was collected from 26 May 2020 to 31 March 2022 containing individuals that tested positive or negative for SARS-CoV-2 [8]. Here, we selected a subpopulation of individuals living in the same household. We included 138 individuals who belonged to 58 different households. Age of the 75 females and 63 males ranged from 1 to 87 years, with a median of 51 years (age distributions are provided as Appendix A). We analyzed these individuals for cytokine polymorphisms of TNF-**α**, IFN-γ, TGF-β1, IL-6, and IL-10. All infected individuals included had moderate disease severity. Only six subjects out of the total cohort of 527 individuals were hospitalized and required non-invasive ventilation.

We grouped the 58 households by the SARS-CoV-2 PCR results into (i) households whose members had contradicting PCR results (PCR+/−, 24 households, 62 individuals; see Figure 1); (ii) households whose members only had PCR positive results (PCR+/+, 29 households, 61 individuals); and (iii) households whose members had PCR-negative results solely (PCR−/−, 5 households, 15 individuals).

From the above-mentioned total cohort of 527 individuals, a subcohort, the cytokine population cohort of 138 individuals, was examined. To avoid an over-representation of specific genotypes due to genetic relationships, we excluded grandparents and grandchildren (Table 1). The remaining 128 individuals represent the population cohort (see Figure 1, left part). For the particular household analysis, we included also related, such as grandparents and grandchildren (see Figure 1, right part).

Informed consent was obtained from all subjects involved in this study by the Ethics Committee of University Leipzig Medical Faculty, 195/20-ek May 2020.

### 2.2. DNA Isolation and Cytokine Genotyping

Genotyping for cytokine polymorphisms was performed for transforming growth factor-β1 (TGF-β1), tumor necrosis factor-α (TNF-α), Interleukin-6 (IL-6), Interleukin-10 (IL-10), and Interferon-γ (IFN-γ) (Table 2). Therefore, DNA was isolated from peripheral blood samples of 138 patients in 58 households. Polymerase chain reaction with sequence-specific primers (PCR-SSP) was performed for genotyping using One Lambda Cytokine Genotyping Tray kit, CYTGEN_004C01, according to the manufacturer’s instructions. For negative control, 1 µL of distilled H_2_O was distributed in the negative control tube along with 9 µL of a mix of 180 µL D-mix provided with the kit and 1 µL of recombinant taq polymerase as recommended by the manufacturer (One Lambda, Inc., West Hills, CA, USA).

For the master mix, 19 µL of DNA (concentration 25–200 ng/µL) was mixed with 180 µL of D-mix included in the kit and 1 µL of recombinant Taq polymerase (5 units/µL). Then, 10 µL of the master mix was distributed to each test tube containing specific primers for tested cytokines in the 96-Well Primer Set tray. For PCR, the GeneAmp© PCR System 9700 was used under the following conditions: initial denaturation step 1 96 °C, 130 s; step 2 63 °C, 60 s; denaturation 96 °C, 10 s; annealing + extension 63 °C, 1 min (9 cycles); denaturation 96 °C, 10 s; annealing 59 °C, 50 s; extension 72 °C, 30 s (20 cycles). PCR products were visualized on 2% agarose gel and photographed using the Vilber E-Box VX2 with UV-radiation. Results were analyzed using the scheme provided by the manufacturer with the kit and re-read by a second trained person from the laboratory.

Manufacturer guidelines associate varying potential of cytokine production to each of the loci (Table 2). This is a surrogate measure reflecting the genetic precondition of a person for lower, intermediate, or higher release of the cytokine. These categories are related to the corresponding serum abundances [34]. NB, we did not measure the cytokine concentration in the serum/plasma of the individuals but analyzed the molecular polymorphism and thus the potential to release the respective cytokine.

### 2.3. Statistical Analysis

For statistical analysis, SPSS 29 (IBM Corp., Armonk, NY, USA) and R 4.2.3 were used. Statistical significance was tested using two-tailed Fisher’s exact test. A value *p* < 0.05 was regarded as significant.

## 3. Results

### 3.1. Cytokine Polymorphisms in the Population Cohort

In the 138 individuals studied, we found different distributions of the cytokine polymorphisms of the T cell Th1- (TNF-α, IFN-γ), Th2- (IL-10) and Treg (TGF-β1) signature cytokines as well as the Treg and Th17 regulatory cytokines (TGF-β1, IL-6).

Looking at all individuals within the whole cohort (grey bars), G/G low was the most prevalent polymorphism for TNF-α with about 75%, with few numbers of T/A high (20%), and A/A high almost non-existent (Figure 2, top left). This G/G polymorphism is associated with low potential for TNF-α release. For IFN-γ, T/A intermediate was the most frequent with 50%; about a third had A/A low, and 20% T/T high. Regarding the innate immunity biomarker and Th17 regulatory cytokine IL-6, G/C high (50%) had the largest share in the cohort, followed by the other high release potential polymorphism G/G high (30%), and just under 20% of individuals had C/C low (Figure 2, top row).

The Th2 pathway cytokines IL-10 and the Treg regulatory cytokine TGF-β1 also showed strong intermediate and high cytokine release polymorphisms: For IL-10, we found GCC/GCC high in 24% of the individuals alongside two common intermediate release potential polymorphisms, GCC/ACC (28%) and GCC/ATA (20%). T/T G/G high (41%) and T/C G/G high (36%) were the predominant polymorphisms for TGF-β1; 13% (T/C G/C) and 7% (C/C G/G) showed intermediate cytokine release potential polymorphisms and just a small portion of 3% C/C G/C low. Four polymorphisms of the cytokine TGF-β1 were tested but not found in the cohort: T/T G/C intermediate, C/C C/C low, T/T C/C low and T/C C/C low. 

As depicted in Figure 2 and looking at noteworthy differences in the distribution of polymorphisms between PCR-positive and PCR-negative individuals within the whole cohort (orange and blue bars), TNF-α looked very homogeneous, whereas the other Th1-pathway cytokine had different results: Regarding IFN-γ, there was an overexpression of T/A intermediate in PCR-negative individuals compared to the PCR-positive group (*p* = 0.07), whereas T/T high (*p* = 0.46) and A/A low (*p* = 0.20) were found more frequent, but not significant, in PCR-positive individuals than in the PCR-negative group. Concerning IL-6, G/G high was more prevalent for negative individuals and found less in PCR-positive individuals (*p* = 0.12), whereas G/C high was more common in PCR-positive individuals (*p* = 0.11; see Figure 2, top right). 

Looking at Th2 cell and Treg cell cytokines in the bottom row, the GCC/GCC high polymorphism (*p* = 0.48) for IL-10 and GCC/ACC intermediate (*p* = 0.51) were found more frequently, but not significant, in negative individuals. In contrast, 12% of PCR-positive individuals in the whole cohort had the ATA/ATA low polymorphism, with no PCR-negative individual with that polymorphism, which was statistically significant (*p* = 0.04).

For TGF-β1, the T/T G/G high polymorphism was slightly more common in PCR-negative individuals. T/C G/G high, on the other hand, was about 15% lower in PCR-negative than in PCR-positive individuals (*p* = 0.10). The two intermediate cytokine release potential polymorphisms for TGF-β1 made up a higher percentage in PCR-negative individuals than in the other groups (Figure 2 middle, top row).

### 3.2. Cytokine Polymorphisms in the Different Household Types

The distribution of cytokine polymorphisms in different SARS-CoV-2 PCR households is shown in Table 2. There was a quite similar distribution of the SARS-CoV-2 PCR+/− households compared to the population cohort of 138 individuals. All polymorphisms we found in the population cohort were also represented in the PCR+/− households.

In the PCR+/− household group, the cytokines TNF-α and IFN-γ show no different distribution of the cytokine polymorphism compared to the population cohort (Table 2 and Figure 2).

While the distribution of IL-6 within PCR-positive individuals for the high release potential polymorphisms was almost the same in PCR+/− and the whole cohort, the distribution in PCR-negative individuals showed a strong shift toward G/C high, up to about 60%, in the PCR+/− group. Being evenly distributed in the whole cohort, IL-6 G/C (high) was more frequent (in 35%) than IL-6 G/G high in PCR-negative individuals (*p* = 0.76) in the PCR+/− group.

The distribution of IL-10 polymorphism showed some differences when comparing the whole cohort and the PCR+/− households. The gap in the frequency of GCC/GCC high (*p* = 0.48) and GCC/ACC intermediate (*p* = 0.40) between positive and negative individuals was larger than in the whole cohort, but not significant. Compared to the whole cohort, it stood out that ACC/ACC low was not found in PCR-negative individuals in the PCR+/− households (*p* = 1.00). IL-10 ACC/ATA (low) was 5% more frequent in PCR-negative individuals in PCR+/− households compared to the PCR-negative individuals in the whole cohort.

For TGF-β1, some polymorphisms were not found in the studied cohort, as indicated in Table 2. The high TGF-β1 release potential polymorphism T/C G/G in the PCR+/− households showed no differences between positive and negative individuals compared to the whole cohort in which the negative individuals had the lowest percentage (23%) of this polymorphism (*p* = 0.41). In the PCR+/− cohort, we observed no low TGF-β1 release potential polymorphism (C/C G/C) within negatively tested individuals compared to 3% in the whole cohort (*p* = 1.00). 

Table 2 also shows the distribution of polymorphisms within households with only SARS-CoV-2-PCR-positive members (PCR+/+). The distribution of the polymorphisms was comparable to that of the individuals within the population cohort (Table 2 and Figure 2).

For TNF-α, G/G low was the predominant polymorphism with 77%; 20% had G/A high and 3% A/A, which was a 2% increase compared to the PCR-positive individuals in the whole cohort. INF-γ showed a similar distribution compared to positive individuals of the whole cohort, with just a 3% shift from T/A intermediate to the A/A low polymorphism.

There was no notable difference in the distribution for IL-6 in these households compared to all positive individuals.

IL-10 GCC/GCC high was about 6% more common in individuals of the PCR+/+ households, whereas both intermediate cytokine release potential polymorphisms GCC/ACC (6%) and GCC/ATA (3%) were less frequent. The low release potential polymorphisms showed the same distribution pattern with a little shift toward ACC/ACC and ATA/ATA of about 1%, respectively 2% (Table 2).

TGF-β1 polymorphisms also looked similar, only with both high release potential polymorphisms T/T G/G and T/C G/G even slightly more pronounced, with a prevalence over 40%. T/C G/G intermediate had a 1% higher proportion, whereas C/C G/G intermediate and C/C G/C low lost about 2% compared to PCR-positive individuals and to the whole cohort.

The distribution patterns of cytokine polymorphisms among the members of all PCR-negative households (PCR−/−) showed some differences compared to the negative individuals within the whole cohort and the members of all positive households (see Table 2). 

The A/A high polymorphism of TNF-α was not found in the PCR−/− households, and G/G low was even more predominant, making up a share of 93%, or a 20% increase compared to negative individuals (*p* = 0.15).

While the T/T high polymorphism of IFN-γ stayed around 10%, the distribution shifted from T/A intermediate to the A/A low polymorphism, which showed a 13% increase compared to PCR negatives in the whole cohort, to 33%.

Larger differences could be seen for IL-6. The high release potential polymorphisms were equally distributed in the whole cohort but showed a strong predominance of the G/G high polymorphism (73%) in the PCR−/− households, which was a 32% increase (*p* = 0.06). G/C high was 20% and C/C 10% less frequent than in PCR-negative individuals within the whole cohort.

For the Th2-pathway cytokine IL-10, the GCC/GCC high polymorphism was not as frequent as in the whole cohort (20%). The intermediate release potential cytokines were still predominant in those household members, but the distribution shifted toward the GCC/ATA polymorphism; it was 13% percent more frequent compared to all negative individuals (*p* = 0.47), making it the most common IL-10 polymorphism in this cohort, with 33%. ACC/ACC low was a little more frequent with 7%, where ACC/ATA low showed the same distribution as in all PCR-negative individuals.

The distribution pattern of TGF-β1 showed quite a few differences compared to all PCR-negative individuals in the whole cohort. While being predominant in the whole cohort, the T/T G/G high polymorphism was 11% less frequent and made up just 33% within the PCR−/− household cohort. The other high release potential polymorphism, T/C G/G, showed a decrease to 7% within the PCR−/− cohort, whereas it was found in 24% of all PCR-negative individuals (*p* = 0.24). T/C G/C intermediate, on the other hand, was the most frequent polymorphism in the PCR−/− households with 40%, a 22% increase compared to all PCR-negative individuals (*p* = 0.15). C/C G/G intermediate and C/C G/C low were both slightly more frequent than in PCR-negative individuals within the whole cohort.

Two of the cytokines examined differed significantly in polymorphisms between the PCR+/+ and PCR−/− households. Expression-relevant polymorphisms of IL-6 and TGF-ß1 showed opposite tendencies (Figure 3). For IL-6, the difference in the distribution of the two polymorphisms with high release potential was obvious. While the G/G polymorphism of the IL-6 promoter was highly prevalent in PCR−/− household members (73%), this polymorphism was only found in 26% of people in PCR+/+ households (significance for the difference: *p* < 0.01). For IL-6 promoter G/C high, household members showed an opposite pattern, with 20% occurrence in PCR−/− households and 57% in PCR+/+ households (*p* = 0.02).

TGF-β1 also showed significant differences for two polymorphisms when comparing PCR+/+ with PCR−/− households. Just 7% of individuals in negative households showed the T/C G/G high polymorphism, compared to 43% in all positive households (*p* = 0.01).

The intermediate release potential of TGF-β1 T/C G/C was higher than in other groups, 40% in PCR−/− households compared to 12% in PCR+/+ households (*p* = 0.02). Compared to individuals from all positive households, these two polymorphisms occurred in reverse patterns.

### 3.3. Inheritance of Cytokine Polymorphisms within Families

The inheritance of the cytokine polymorphisms, which embody the potential for cytokine release, follows the inheritance rules described by Mendel [35]. The genes of the cytokines studied here are located on different chromosomes (TNF-α: chromosome 6, TGF-β1: chromosome 2, IL-10: chromosome 1, IL-6: chromosome 7, and IFN-γ: chromosome 12). Therefore, they are not linked. In our cohort, we identified a total of 8 “complete” families (Table 3 and Appendix A). We term families consisting of mother, father and at least one child as “complete” families. For better visualization, the respective paternal and maternal inherited sequences are given in colors (Table 3). Among informative genotypes, like for IL-10, the paternal genotype ATA (blue) was inherited by both children, while the other genotype GCC (yellow) was not inherited. Similarly, the paternal genotype ACC (black) was inherited by the first child, while the second genotype GCC (red) was inherited by the second child. In this way, we analyzed the eight families as reported in the Appendix A.

The results depicted here show the inheritance of the cytokine release potential in these families.

## 4. Discussion

This study offers insight into relevant cytokine polymorphisms within a Caucasian cohort in Saxony regarding susceptibility to SARS-CoV-2 infection. The distribution of polymorphisms is consistent with pre-existing data from Germany (Appendix A). Compared to cohorts from other parts of the world, there were some notable differences in the distribution of cytokine polymorphisms, like those that are known from studies on other immunogenetic markers like HLA [7,8]. This might be due to an evolutionary pressure of different environments and their specific pathogens. With regard to the different CD4+ cell populations, the role of helminth infections was previously discussed [36], Th2 being the leading pathway for the immune defense against helminths and other protozoa [14,20]. With less and less protozoal diseases in high-resource countries, an evolving disbalance between Th1 and Th2-pathways may occur and, together with other regulatory CD4+ T cell populations, might have an influence on the host immune response to SARS-CoV-2 and the subsequent organ and tissue damage caused by a hyperinflammatory state, characterized by a cytokine storm in the late phase of COVID-19 [36].

Furthermore, while the Th2 pathway is thought to be associated with IgE-mediated allergy [37], Th17 and especially Treg cells maintain tolerance and control of unwanted autoimmunity. The extent to which allergies played a role in the individuals in our cohort cannot be answered, as this was not the aim of the study and, therefore, was not recorded.

Furthermore, we found some interesting conspicuities regarding particular polymorphisms: In COVID-19, IFN-γ, as one of the most important cytokines for the host’s adaptive immune defense [28], is initially measured in high concentrations in the blood of individuals with few or only mild symptoms, whereas persistent high blood concentrations can be indicative of a severe course [27,28]. Although not statistically significant, we observed more polymorphisms that resulted in intermediate cytokine release potential of IFN-γ in the PCR-negative individuals. Both high and low cytokine release potentials were rarer than in the PCR-positive individuals. Further studies focusing on comparisons between the release potentials and protein concentration locally at the site of infection and systemically in the blood may elucidate the significance of these results. Some studies suggest that IFN-γ and TGF-β1 are negatively correlated to COVID-19 patients, with TGF-β1 being released form the respiratory epithelium early in the course of infection, which then regulates the release of IFN-γ and may trigger an excessive immune response [29]. The release potentials for both cytokines were at an intermediate level in our cohort. In contrast, previous studies showed increased TGF-β1 blood concentrations in COVID-19 [30] and suggested an involvement of this cytokine in the early pathogenesis of the disease and the development of clinical sequelae [29,30] such as extracellular matrix degradation and pulmonary fibrosis as serious consequences of COVID-19.

Therapeutic interventions targeting TGF-β1 are already discussed [29] as well as implications of the IFN-γ action. Significant differences between PCR+/+ and PCR−/− households in our study for the high and intermediate release potentials as a genetic variation for this important cytokine may play a role in the susceptibility to SARS-CoV-2 and should be subject in further studies with more participants.

Furthermore, significant differences were found within the IL-6 cytokine polymorphisms. This cytokine plays a pivotal role in the pathogenesis of COVID-19, shows significantly elevated blood levels in patients [26,27] and can be used as a predictive marker for the course of the disease [31]. The anti-IL-6 antibody Tocilizumab is used in critically ill patients to mitigate symptoms and improve the outcome [29,31,32,33,34,35,36,37,38,39]. With these cytokines being the subject of many studies with severe COVID-19 patients and different approaches of therapeutic options [9,26,31,32,38], genetic polymorphisms could be investigated in patients at different stages of the disease to evaluate the impact of these cytokine polymorphisms on the severity of COVID-19 at the genetic level [40,41,42]. For both TGF-β1 and IL-6 data implying an important role for the course and the outcomes of COVID-19 and may do so concerning the susceptibility. TGF-β1, a key cytokine in the present study, is also associated with regulatory T cells. These downregulate the pro-inflammatory pathways Th1 and Th17 and thus also increase a Th2-triggerd response. The latter was evident in those in the PCR− group. The initial dose leading to a more Th1-dominant inflammatory response [39]—the reduced expression of ACE2 in the cilia-bearing respiratory epithelia and IL-13-dependent Th2-inflammatory mechanisms in distinct IL-13-repsonsive respiratory epithelia cells in subjects with allergies [43]—could play a role.

A low release potential of cytokines can lead to reduced transcription and protein concentrations. In our population, we observed markers of higher release potentials in the pro-inflammatory cytokines such as the Th1 IFN-γ and TNF-α, as well as in the pleiotropic cytokine IL-6 (Table 4). All PCR−/− individuals tended to have a higher number of the low cytokine polymorphism. These data may indicate that a more regulated and less inflammatory response confers a higher chance of resilience to SARS-CoV-2.

In our study, we focused on the model of CD4+ T cell differentiation with the different subpopulations (Th1, Th2, Th17, and Treg) and examined known genetic polymorphisms for the cytokines that are important for this. Especially, the Th1/Th2-pathway model can be used as a common model for understanding the host response to different pathogens [14,20,44] and also COVID-19 [26,36,45]. With regard to the other, more regulatory subpopulations, this concept could be too simple for several diseases [46] and possibly also inadequate for the immunological processes in SARS-CoV-2 infection [15,47,48]. In particular, the role of Th17 cells in the course of the disease with development of a cytokine storm in severe illness is already discussed [15,27,47,48]. Our results point to a possibly important role of differentiation-inducing cytokines (TGF-β1, IL-6) for Th17 and Treg cells, which could usefully expand the CD4+ T cell-driven orchestration of the adaptive immune response in the course of disease and the mechanisms of inherited resistance to SARS-CoV-2 infection.

## 5. Conclusions

Our data suggest that genetically determined changes in cytokine expression, which play an essential role in the function and regulation of different CD4+ T cell subpopulations, may have an influence on susceptibility to SARS-CoV-2 infection. In particular, TGF-β1 appears to play a role in facilitating transmission, while IL-6-associated polymorphisms, which may lead to higher levels of expression, play an opposite role. This suggests that distinct regulatory T cell pathways that are driven by Treg and Th17 cells are associated with increased susceptibility to SARS-CoV-2 infection.

As this study was performed in white Caucasian Europeans, it would be important to repeat the study in different populations, to generalize these findings.

## Figures and Tables

**Figure 1 biology-12-01385-f001:**
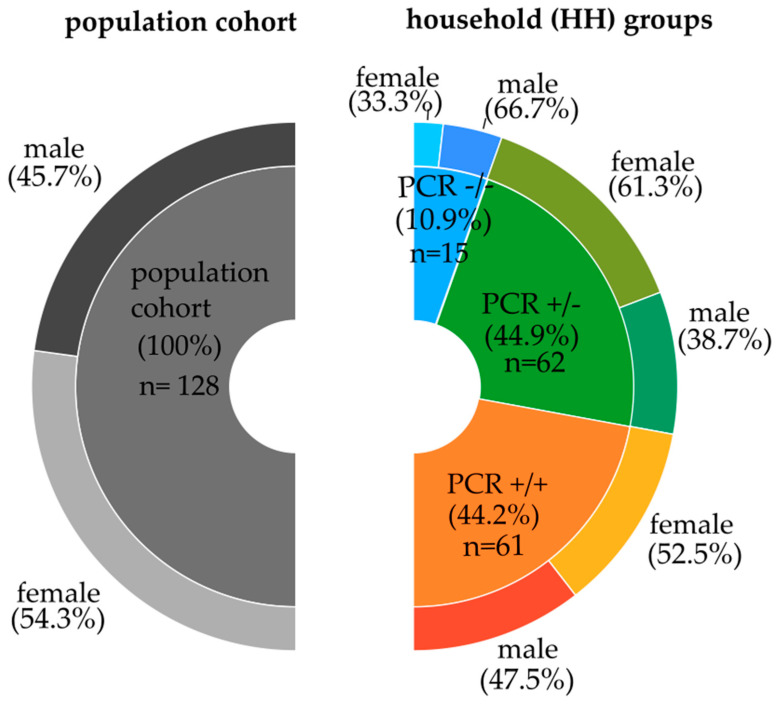
Gender distribution in the Saxon population and the different household groups. The left part depicts the population cohort, excluding grandparents and grandchildren; the right part represents the household groups according to the SARS-CoV-2 PCR characteristics, respectively, including grandparents and grandchildren.

**Figure 2 biology-12-01385-f002:**
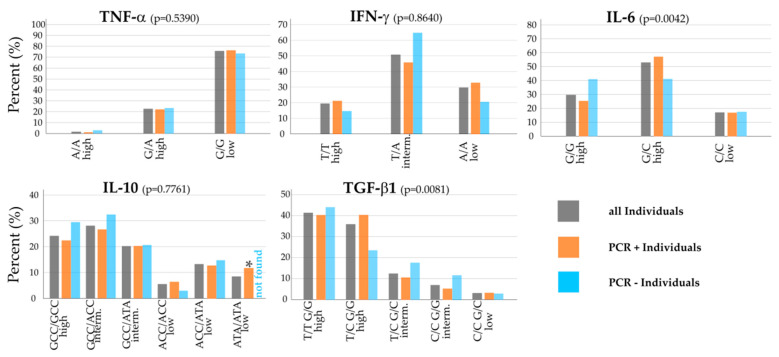
Distribution of the cytokine polymorphisms within the Saxon population cohort. Grey bars represent the percentage of the particular polymorphism within all individuals, orange bars represent the percentage within SARS-CoV-2 positive individuals, blue bars represent the percentage within SARS-CoV-2 negative individuals in the cohort. The top row shows the tested Th1 cytokines (TNF-α, IFN-γ), which are considered as pro-inflammatory, and the regulatory cytokine (IL-6). The bottom row shows the Th2 cytokine IL-10, which is considered as anti-inflammatory, and the regulatory cytokine TGF-β1. For TGF-β1, some polymorphisms (intermediate T/T G/C, low C/C C/C, low T/T C/C and low T/C C/C) were not found in the cohort. * indicates a significance of *p* < 0.05 between ATA/ATA low polymorphism PCR+ and PCR− individuals.

**Figure 3 biology-12-01385-f003:**
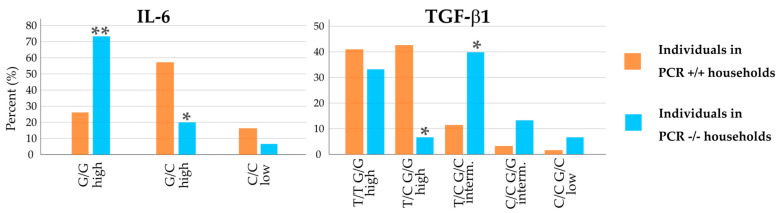
Distribution of individuals in PCR+/+ households (orange bars) and PCR−/− households (blue bars). interm. = intermediate. * indicates significance with *p* < 0.05; ** indicates significance with *p* < 0.01 for difference in distribution in PCR−/− households compared to PCR+/+ households, performed by two-tailed Fisher’s exact test.

**Table 1 biology-12-01385-t001:** Genetic relations of individuals within the particular households.

Genetic Relation ^1^	Whole Cohort (%)	Subgroup PCR +/− (%)	Subgroup PCR +/+ (%)	Subgroup PCR −/− (%)
F	7.2	8.1	4.9	13.3
FM	0.7	1.6	0	0
MM	0.7	1.6	0	0
M	11.6	11.3	11.5	13.3
F2	0.7	1.6	0	0
0	60.9	51.6	73.8	46.7
C1	11.6	12.9	8.2	13.3
C2	4.3	6.5	0	13.3
C1C1	1.4	3.2	1.6	0
C1F2	0.7	1.6	0	0
total	100	100	100	100

^1^ F = father, M = mother, MM = mother of mother, FM = father of mother, F2 = father two (father of different child), C1 = child 1, C2 = child 2, C1C1 = child of child, C1F2 = child of father two (with mothers of household).

**Table 2 biology-12-01385-t002:** Distribution of the cytokine polymorphisms within the PCR+/−, PCR+/+ and PCR−/− households.

Cytokine	Polymorphism	Cytokine Release Potential	PCR+/− Households (%)	PCR+/+ Households (%)	PCR−/− Households (%)
TNF-α (promoter -308G, -308A)	G/G	low	75.8	77.0	93.3
	G/A	high	22.5	19.7	6.7
	A/A	high	1.6	3.3	not found
IFN-γ (intron +874T, +874A)	T/T	high	27.4	21.3	13.3
	T/A	intermediate	50	47.5	53.3
	A/A	low	22.6	31.1	33.3
IL-6 (promoter -174C, -174G)	G/G	high	24.2	26.2	73.3
	G/C	high	54.8	57.4	20.0
	C/C	low	21	16.4	6.7
IL-10 (promoter -1082A, -1082G, -819T, -819C, -592A, -592C)	GCC/GCC	high	19.3	27.9	20.0
	GCC/ACC	intermediate	29.0	21.3	26.7
	GCC/ATA	intermediate	22.6	23.0	33.3
	ACC/ACC	low	4.8	4.9	6.7
	ACC/ATA	low	16.1	13.1	13.3
	ATA/ATA	low	8.1	9.8	not found
TGF-β1 (codon 10T, 10C, 25C, 25G)	T/T G/G	high	38.7	41.0	33.3
	T/C G/G	high	37.1	42.6	6.7
	T/C G/C	intermediate	9.7	11.5	40.0
	C/C G/G	intermediate	11.3	3.3	13.3
	T/T G/C	intermediate	not found	not found	not found
	C/C G/C	low	5.2	1.6	6.7
	C/C C/C	low	not found	not found	not found
	T/T C/C	low	not found	not found	not found
	T/C C/C	low	not found	not found	not found

**Table 3 biology-12-01385-t003:** Example of the inheritance of the cytokine polymorphism in a PCR+/− household.

Household Number	Genetic Relation	PCR Household	SARS-CoV-2 PCR	TNF-α	Cytokine Release Potential TNF-α	TGF-β1	Cytokine Release Potential TGF-β1	IL-10	Cytokine Release Potential IL-10	IL-6	Cytokine Release Potential IL-6	IFN-γ	Cytokine Release Potential IFN-γ
LEI_042	M	+/− PCR	positive	G/A ^1^	high	T/C G/G	high	GCC/ATA	intermediate	G/G	high	A/A	low
LEI_042	F	+/− PCR	negative	G/G ^1^	low	C/C G/G	intermediate	GCC/ACC	intermediate	G/C	high	T/A	intermediate
LEI_042	C1	+/− PCR	negative	G/G ^2^	low	C/C G/G	intermediate	ACC/ATA	low	G/G	high	A/A	low
LEI_042	C2	+/− PCR	positive	G/G ^2^	low	C/C G/G	intermediate	GCC/ATA	intermediate	G/C	high	T/A	intermediate

^1^ Blue and yellow colors indicate the alleles of the mother (M), red and black the alleles of the father (F). ^2^ Alleles of the children are assigned as inherited from mother or father, respectively, as indicated by the colors.

**Table 4 biology-12-01385-t004:** Inheritance of cytokine polymorphism in “complete” families ^1^.

Cytokine	Polymorphism	Cytokine Release Potential	Prevalence in PCR Positive Individuals (%)	Prevalence in PCR Negative Individuals (%)
TNF-α (promoter -308G, -308A)	G/G	low	94.11	100
	G/A	high	5.88	not found
	A/A	high	not found	not found
IFN-γ (intron +874T, +874A)	T/T	high	35.29	23.52
	T/A	intermediate	41.18	52.94
	A/A	low	23.52	23.52
IL-6 (promoter -174C, -174G)	G/G	high	35.29	52.94
	G/C	high	64.7	47.05
	C/C	low	not found	not found
IL-10 (promoter -1082A, -1082G, -819T, -819C, -592A, -592C)	GCC/GCC	high	11.76	17.65
	GCC/ACC	intermediate	29.41	23.52
	GCC/ATA	intermediate	17.65	35.29
	ACC/ACC	low	11.76	not found
	ACC/ATA	low	11.76	23.52
	ATA/ATA	low	17.65	not found
TGF-β1 TGF-β1 (codon 10T, 10C, 25C, 25G)	T/T G/G	high	41.18	17.65
	T/C G/G	high	41.18	23.52
	T/C G/C	intermediate	11.76	35.29
	C/C G/G	intermediate	5.88	23.52
	T/T G/C	intermediate	not found	not found
	C/C G/C	low	not found	not found
	C/C C/C	low	not found	not found
	T/T C/C	low	not found	not found
	T/C C/C	low	not found	not found

^1^ Families consisting of mother, father, and at least one child in common.

## Data Availability

The data supporting this study’s findings are available from the corresponding author upon reasonable request.

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
