# Peer review of "Genetic Predisposition to SARS-CoV-2 Infection: Cytokine Polymorphism and Disease Transmission within Households"

_biology, 2023, doi:10.3390/biology12111385_

Round 1

Reviewer 1 Report

Comments and Suggestions for Authors

This manuscript describes  a cohort study on cytokine polymorphism and SARS-CoV-2 infection in different households, indicate that the balance of the Th1 and Th2 cytokines which is genetically determined may play a pivotal role in the transmission of and susceptibility to SARS-CoV-2 infection, and suggests a higher Th2-pathway activity leading to increased susceptibility for SARS-CoV-2 infection. It is known that the Th2-pathway activity is important for IgE-mediated allergy, wether the authors have any evidence demonstrated that patients with allergies are susceptible to SARS-CoV-2 infection? I wonder how the higher Th2-pathway activity can involve in the transmission of and susceptibility to SARS-CoV-2 infection. Moreover, TGF-b is known that associates with Treg, why authors didn't discuss on Treg? 

Comments on the Quality of English Language

need native edting

Reviewer 2 Report

Comments and Suggestions for Authors

Authors have addressed in interesting issue looking for gene polymorphisms differences that can explain the great variability of susceptibility to SARS-CoV2 infection. They used an intriguing model to unravel significant differences in subjects, who undergo infected or not and sharing the same environment with infected indiviaduals.

However, data are poorly presented and need to be better analyzed.

- first of all, data should be better analyzed from a statistic point of view: too many comparisons were made using univariate tests. An approach taking it into account should be attempted (i.e. multivariate analysis and post-hoc correction);

- also for continuous variables (citokine levels) no adeguate analysis was provided (i.e. linear multiple regression);

- as the Authors stated in the introduction section, many factors (age, diseases, etc) play a role in the susceptibility to the infection. No correction for confounding factors was done;

- was the Hardy-Weinberg equilibrium preserved?

- data should be analyzed and presented comparing different settings (households ++ vs. +- vs. --) and different genetic models should be tested (dominant, recessive, co-dominanat).

Round 2

Reviewer 1 Report

Comments and Suggestions for Authors

In this revised manuscript, authors have improved the discussions on Th17 and Treg. Authors did not measure the cytokine concentration in the serum/plasma of the individuals, I wonder whether the cytokine release potential is consided that equals cytokine concentrations. Also, the definition and significance of high and low cytokine release potential remains unclear. In this study, TGF-b and IL-6 polymorphism were significantly different between PCR positive and negative individuals in households, which genetic alteration is crucial for cytokine release and also critical for the susceptibility to infection and severity of COVID-19.

Comments on the Quality of English Language

please check by native speaker

Author Response

Thanks for the comment to the reviewer:

Indeed we did not measure cytokine concentrations in the serum or plasma as stated in the text. However, we included literature in the text line 134 [24, 25] and in line 496 [40-42] were the release potential is described. Our manuscript has been checked by an english speaking colleague.

Reviewer 2 Report

Comments and Suggestions for Authors

None.

Author Response

Thanks to the reviewer